# Metatranscriptomic Comparison of Viromes in Endemic and Introduced Passerines in New Zealand

**DOI:** 10.3390/v14071364

**Published:** 2022-06-23

**Authors:** Rebecca K. French, Antoine Filion, Chris N. Niebuhr, Edward C. Holmes

**Affiliations:** 1Sydney Institute for Infectious Diseases, School of Life and Environmental Sciences and Sydney Medical School, The University of Sydney, Sydney, NSW 2006, Australia; rebecca.french@sydney.edu.au; 2Department of Zoology, University of Otago, 340 Great King St., Dunedin 9016, New Zealand; aflion90@gmail.com; 3Manaaki Whenua–Landcare Research, P.O. Box 69040, Lincoln 7640, New Zealand; niebuhrc@landcareresearch.co.nz

**Keywords:** metagenomics, Passeriformes, siadenovirus, iltovirus, avastrovirus

## Abstract

New Zealand/Aotearoa has many endemic passerine birds vulnerable to emerging infectious diseases. Yet little is known about viruses in passerines, and in some countries, including New Zealand, the virome of wild passerines has been only scarcely researched. Using metatranscriptomic sequencing we characterised the virome of New Zealand endemic and introduced species of passerine. Accordingly, we identified 34 possible avian viruses from cloacal swabs of 12 endemic and introduced bird species not showing signs of disease. These included a novel siadenovirus, iltovirus, and avastrovirus in the Eurasian blackbird (*Turdus merula,* an introduced species), song thrush (*Turdus philomelos,* introduced) and silvereye/tauhou (*Zosterops lateralis*, introduced), respectively. This is the first time novel viruses from these genera have been identified in New Zealand, likely reflecting prior undersampling. It also represents the first identification of an iltovirus and siadenovirus in blackbirds and thrushes globally. These three viruses were only found in introduced species and may pose a risk to endemic species if they were to jump species boundaries, particularly the iltoviruses and siadenoviruses that have a prior history of disease associations. Further virus study and surveillance are needed in New Zealand avifauna, particularly in *Turdus* populations and endemic species.

## 1. Introduction

Isolated from the rest of the world for more than 50 million years, New Zealand’s avifauna evolved in a unique ecosystem absent of land mammals, and as a result, are highly taxonomically and ecologically distinct [1,2,3]. However, deforestation and the introduction of multiple mammalian predators and non-native bird species [4] have resulted in a vastly altered avifaunal community, including the loss of half the native bird taxa [5]. Today, many ecological communities are dominated by introduced bird species that are better adapted to the changed environment and the presence of mammalian predators [4,6,7]. The introduced bird species include Eurasian blackbirds (*Turdus merula*) which were first introduced to New Zealand in 1862, and song thrushes (*Turdus philomelos*) in 1867 by acclimatisation societies aiming to establish exotic species in New Zealand [8]. Silvereyes/tauhou (*Zosterops lateralis)*, however, are thought to have naturally dispersed to New Zealand from Australia in the early 1800s [9]. Many of the endemic species still extant are at risk of extinction with small population sizes and low genetic diversity, making them particularly vulnerable to disease outbreaks from emerging pathogens. In addition, their small population size makes them vulnerable to stochastic events and may also mean a limited immune repertoire. For example, a severely bottlenecked population of the South Island robin/kakaruai *Petroica australis* had a significantly weaker T-cell-mediated immune response than a more genetically diverse population [10]. The isolated evolution of these endemic species may also make them particularly vulnerable to the introduction of exotic pathogens from overseas [11], yet for some species being relatively closely related to introduced species (at the order level) may also increase the chances of cross-species virus transmission [12].

Despite the clear threat of emerging infectious disease, relatively little is known about the virome of New Zealand endemic or introduced birds. There has been no research on the viruses currently circulating in introduced species and the threat these may pose to the more vulnerable endemic species if cross-species transmission were to occur. Similarly, there is no regular monitoring of most endemic species nor ongoing surveillance. For example, an arthropod-borne virus (arbovirus) was detected in native and introduced passerines (Aves: Passeriformes) on the west coast of New Zealand in 1962, with the Eurasian blackbird and song thrush thought to be reservoirs [13]. However, despite concerns about the threat to native species (and possible links to human outbreaks of influenza-like disease) there was no surveillance for four decades after the initial discovery, while a serological study in 2006/2007 indicated the virus was still present in reservoir species [14]. Despite the limited research, it is thought that cross-species virus transmission has occurred between introduced and native bird species to the detriment of the native populations. For example, the prevalent *Avipoxvirus* strain (species *fowlpox virus,* HPB strain, subclade A1) was likely brought to New Zealand through introduced avian hosts [15], and may be a cause of ongoing mortalities in endangered black robin/karure (*Petroica traversi*) and shore plover/tuturuatu (*Thinornis novaeseelandiae*) populations [16]. This strain and others present in New Zealand may have been introduced along with exotic passerines purposely brought over from Europe [15].

Passeriformes is the most specious avian order, constituting 60% of avian diversity [17], and therefore they may also be expected to harbour a high diversity of viruses. In the Northern Hemisphere, passerines are known to play important roles in infectious disease emergence and spread. In Europe they have a high prevalence of arboviruses, commonly acting as amplifying hosts [18,19], while in North America they have played an important role in the establishment and spread of the West Nile virus [20]. However, the virome of wild passerines has received little research attention, with the first metagenomic analysis published in 2019 and 2020 from samples collected in French Guiana and Spain [21,22,23], followed by a study from China in 2022 [24]. Multiple highly divergent viruses were identified in these studies, including new groups of avastroviruses and a highly divergent gyrovirus. A small metagenomic study of faecal DNA from South Island robins/kakaruai in New Zealand also found a high diversity of DNA viruses [25]. However, it is evident that passerines have been hugely under-sampled globally, including in New Zealand.

Due to recent advancements in RNA sequencing, characterisation of the entire virome of a sample is now achievable using metatranscriptomic (i.e., total RNA) sequencing, capturing both RNA and DNA viruses (via transcribed mRNA). This method enables the comparison of viral abundance and diversity between groups (e.g., between animal populations or species), which was previously not possible on a large scale [26,27]. Herein, we aimed to characterise and compare the virome of New Zealand introduced and endemic species of passerines, identify novel viruses, and discuss the potential for cross-species transmission and disease emergence.

## 2. Materials and Methods

Fieldwork was undertaken between the 28th of November and the 7th of December 2020. Samples were collected from seven different sites in the South Island of New Zealand (Figure 1). Sites were in a variety of predominately native New Zealand bush habitats, including beech forest, lowland podocarp forest and scrub (Appendix A).

Birds were caught using low canopy mist nets. None of the birds showed obvious clinical signs of disease. A cloacal swab was taken (sterile Dryswab™ with fine tip rayon bud, bioMérieux), then cut using scissors sterilised with 70% ethanol, and placed into a tube with 1 mL of RNAlater. Samples were kept cold (6 °C or less) for the duration of the fieldwork using dry ice and a portable fridge/freezer, and stored at −80 °C until processing.

RNA was extracted using the RNeasy plus mini extraction kit (Qiagen) and QIAshredders (Qiagen). The tube containing the swab in RNAlater was thawed and the swab removed from the tube using sterile forceps and placed in 600 µL of extraction buffer. The swab and buffer were vortexed for two minutes at maximum speed. The swab and buffer were then placed into a QIAshredder and centrifuged for five minutes at maximum speed. The flowthrough was retained (avoiding the cell debris pellet) and used in the extraction following the standard protocol in the kit. The RNA was eluted into 50 µL of sterile water.

Extractions were pooled by host species for sequencing (Table 1). In each pool 25 µL of each extraction was used, and this was concentrated using the NucleoSpin RNA Clean-up XS, Micro kit for RNA clean up and concentration (Machery-Nagel). The concentrated RNA was eluted into 20 µL of sterile water.

cDNA libraries were prepared using the Stranded Total RNA Prep with Ribo-Zero Plus (Illumina) that includes Human, Mouse, Rat, Bacteria, and Globin RNA depletion. Libraries were sequenced on the Illumina Novaseq platform (100 bp, 50 M reads per sample, paired end sequencing). Two lanes were used, each sample was sequenced once on each lane and the reads were combined from both lanes for each sample. The corresponding sequencing data have been deposited in the Sequence Read Archive (SRA) with the following accession numbers SAMN27393654-65. The consensus sequences of all novel viruses have been submitted to GenBank and assigned accession numbers ON304002-41.

Nextera paired-end adapters were trimmed using Trimmomatic (0.38) [28]. Bases below a quality of 5 were trimmed using a sliding window approach with a window size of 4. Bases at the beginning and end of the reads were excluded if below a quality score of 3. Sequences below an average quality of ten or less than 100 nucleotides in length were removed using bbduk in BBtools (bbmap 37.98) [29].

Reads were assembled de novo using Trinity (2.8.6) [30]. Blastn (blast + 2.9.0) and Diamond Blastx (Diamond 2.0.9) were used to identify viruses by comparing the assembled contigs to the NCBI nucleotide database (nt) and non-redundant protein database (nr) [31,32]. Contigs with hits to viruses and with an open reading frame greater than 300 nucleotides (for nr hits) were retained. To avoid false-positives, sequence similarity cut-off values of 1E-5 and 1E-10 were used for the nt and nr databases, respectively. Virus abundance was estimated using Bowtie2 (2.2.5) [33]. To account for differences in read depth between libraries, abundance was expressed as the number of reads per million (read count divided by the total number of reads in the library, multiplied by one million). Viruses were assumed to be contamination due to index-hopping from another library if the total read count was <0.1% of the read count in the other library, and they were >99% identical at the nucleic acid level. No viruses were identified that met this criterion. Eukaryotic and bacterial diversity was characterized using CCMetagen (v 1.2.4) and the NCBI nucleotide database (nt) [34,35].

Alpha diversity (i.e., diversity within each sample) was analysed using richness (number of viral species) and both the Shannon and Simpson indices. For this analysis, the entire virome was used, including viruses likely to be of dietary origin. Both the Shannon and Simpson indices measure the richness and evenness of the community, although the Shannon index puts more emphasis on richness, and the Simpson on evenness [36]. Linear models and ANOVAs were used to test for differences in these indices across groups, and generalized linear models were used to test for differences in richness (count data, Poisson distribution). Using a negative binomial distribution to account for overdispersion did not significantly change the results. Using Phyloseq (v1.34.0) in R (v 4.0.5), beta diversity (i.e., shared diversity across host groups) was analysed at the virus species, genus and family level using Adonis tests (permutational multivariate analysis of variance using the Bray–Curtis dissimilarity matrix) and visualised using non-metric multidimensional scaling with a Bray–Curtis dissimilarity matrix, presented as an ordination plot [37,38].

Phylogenetic trees were estimated for those viruses identified from viral families that infect vertebrates. Amino acid sequences were aligned using the E-INS-i or L-INS-i algorithms in MAFFT (7.402) [39] and trimmed using Trimal (1.4.1) [40] with either the automated settings, or with a gap threshold of 0.9 and at least 20% of the sequence conserved. The algorithm and trimming settings for each alignment were chosen based on alignment length and percentage pairwise identity (Appendix A). Individual maximum likelihood phylogenetic trees for each virus family were then estimated using IQ-TREE (1.6.12) [41], with the best-fit substitution model determined by the program and employing the approximate likelihood ratio test with 1000 replicates to assess node robustness. Any sequences used in the phylogenetic analysis with >95% amino acid similarity to each other or known species were assumed to represent the same virus species, with only one representative of each then included in the phylogenetic analysis. Reducing the amino acid similarity cut-off to 90% did not change the number of virus species identified. APE (5.4) and ggtree (2.4.1) were used to visualize the phylogenetic trees and produce figures [42,43]. The map in Figure 1 was created using ggplot2 [44], sf [45], ggspatial [46] and rnaturalearth [47] in R.

Any virus found in the blank negative control libraries (i.e., a sterile water and reagent mix) was assumed to have resulted from contamination likely associated with laboratory reagents. Accordingly, these viruses were removed from the bird libraries and excluded from all analyses. Additionally, any viruses that fell into the same clades as those found in blank libraries were conservatively assumed to be contaminants [48] and similarly removed. As a result of this screen, one circo-like virus in *Circoviridae* (single-strand DNA viruses) was removed from the data set.

## 3. Results

We took cloacal swabs from 59 individual birds of 12 New Zealand passerine species (Table 1). We characterized the RNA viromes using total RNA sequencing and metatranscriptomic analysis.

The 12 sequencing libraries generated had an average of 50.8 million reads per library, and on average 9.7% of these were identified as being derived from viruses (Figure 2). This is a relatively high percentage compared to other studies of cloacal swabs using similar metatranscriptomic techniques [23,49,50]. Analysis of eukaryotic and bacterial diversity showed that all samples primarily comprised eukaryote RNA (accounting for 70–99.6% of assembled contigs) followed by bacteria (0.4–30%). Within the eukaryotic RNA, there was a large variation in the amount of bird RNA, from 2% (grey warbler) to 98% (Eurasian blackbird), but on average the percentage of bird RNA was high (68%). The bacterial taxa varied between samples but commonly included genera from Pseudomonadota, including *Escherichia, Shigella, Pseudomonas,* and *Salmonella.*

### 3.1. Viral Diversity

There was considerable variation in the viral diversity in each library (Table 2), with the tūī having the lowest richness (5 viral species), and the Eurasian blackbird the highest (176 species). Linear models of the richness, Shannon and Simpson indices showed that insectivorous birds had significantly higher viral diversity than birds with other dietary types (richness: deviance = −45.6, *p* = 0.0001, Shannon: F = 17.4, *p* = 0.003, Simpson: F = 20, *p* = 0.002, Figure 3). The richness (number of species) also increased significantly as the total number of reads per library increased (deviance = −267, *p* < 0.0001). However, these trends need to be verified using larger sample sizes, particularly for the herbivorous species as we only sampled two from this group.

Permutational Multivariate Analysis of Variance (PERMANOVA) using the Bray–Curtis dissimilarity matrix revealed that there were no significant differences between viral communities at the viral family, genus or species level according to host species origin, host diet, the number of samples per library, host family and host genus (Adonis test; S3). The average weight of the bird host was significantly associated with viral community composition but only at the viral genus level (F = 1.49, *p* = 0.04), and with a low r-squared value (0.13). Similarly, the total number of reads was significantly associated with viral community composition, but only at the viral species level (F = 1.69, *p* = 0.03), and again with a low r-squared value (0.14). A non-metric multi-dimensional scaling plot also displayed weak clustering according to the total number of reads at the viral species level, with the South Island robin, dunnock and Eurasian blackbird libraries (which each have >90 million reads) forming a loose cluster, and the tomtit, fantail, silvereye, redpoll, chaffinch, and tūī (<50 million reads) similarly forming a loose cluster (Figure 4). At the viral genus level, there was little/no clustering according to host weight (Figure 4). Again, these trends need to be verified using studies with larger sample sizes.

### 3.2. Phylogenetic Analysis of Putative Avian Viruses

Overall, we identified 470 different viruses from 43 different viral families. Viruses belonging to families that exclusively infect plants, fungi, or microorganisms (as identified in previous studies) were assumed to be of dietary origin and excluded from further analysis (436 viruses, 36 families). Phylogenetic trees were generated for 34 viruses from the remaining seven families—the *Adenoviridae, Astroviridae, Caliciviridae, Hepeviridae, Herpesviridae, Picornaviridae* and *Reoviridae* to infer evolutionary relationships (Figure 5, Figure 6, Figure 7, Figure 8, Figure 9, Figure 10 and Figure 11).

#### 3.2.1. DNA Viruses

The *Adenoviridae* is a family of double-stranded DNA viruses that infect vertebrates. We identified one novel adenovirus in the Eurasian blackbird library (termed blackbird siadenovirus), that is most closely related to myna siadenovirus 1 from Australia (60% amino acid similarity in the DNA polymerase gene) (Figure 5). We identified nine contigs from the Eurasian blackbird library belonging to this virus, including a partial DNA polymerase gene (130 amino acids) that was used to estimate a phylogenetic tree. This reveals that the blackbird siadenovirus fits into a well-supported passerine clade within the genus *Siadenovirus* which almost exclusively infects birds, strongly suggesting this is a *bona fide* blackbird virus rather than of dietary origin.

The *Herpesviridae* is a family of double-stranded DNA viruses that infect vertebrates. We found a large number of contigs (*n* = 37) at high abundance (Table 3) in the song thrush library from a novel alphaherpesvirus, including the entire major capsid gene (1459 amino acids). This capsid gene was used to estimate a phylogenetic tree, as the partial DNA polymerase gene was too short (33 amino acids). The novel virus (denoted turdid alphaherpesvirus 1) falls in the genus *Iltovirus* within the *Alphaherpesvirinae*, and is most closely related to psittacid alphaherpesvirus 1 (81.5% amino acid identity, Figure 6). Iltoviruses only infect birds, strongly suggesting turdid alphaherpesvirus 1 infects thrushes, rather than being of dietary origin.

#### 3.2.2. RNA Viruses

The *Astroviridae* is a family of single-stranded positive-sense RNA viruses that infects mainly mammals and birds, but also other vertebrates [51]. We identified three novel astroviruses in our passerine samples. A partial RNA-dependant RNA polymerase contig (211 amino acids) belonging to a novel virus that we have tentatively named avian astrovirus 2 was found in the silvereye library, and was most closely related (88% amino acid similarity) to avian astrovirus from a white-eye bird sample from Hong Kong (Figure 7). Avian astrovirus 2 falls within the genus *Avastrovirus* associated with birds, strongly suggesting that this virus was infecting the sampled silvereyes, rather than being from their diet or microbiome. We also identified two viruses that fell within the currently unclassified bastrovirus clade in the tomtit and song thrush library (Figure 7). These were most closely related to spangled perch bastrovirus (68% amino acid similarity). However, as the likely host of these viruses is unknown it is possible they are of dietary origin. Interestingly, we also found Nelson astrovirus-like 1 (with 99.6% amino acid similarity) in both the robin and tūī libraries. Nelson astrovirus-like 1 was first described in the common wasp in Nelson, New Zealand [52], suggesting that this virus is an invertebrate virus derived from the sampled bird’s diet. Nelson astrovirus-like 1 in the robin and avian associated bastrovirus 2 in the song thrush were found in very high abundances (>4000 reads per million, Table 3).

The *Caliciviridae* is a family of positive-sense, single-stranded RNA viruses, that infects vertebrates. However, a recent metagenomic study identified divergent caliciviruses in the western honey bee *Apis mellifera* (PNG bee virus 1, 12 and 9) [53], suggesting they might also be able to infect invertebrates. We detected four novel species of calicivirus that fall into a clade with PNG bee virus 1 and 12, and “calicivirus mystacina”, a calicivirus found in the lesser short-tailed bat/pekapeka *Mystacina tuberculata* in New Zealand [54] (Figure 8). The presence of the bee viruses in this clade raises the possibility that these are all invertebrate viruses, and both our avian viruses and the bat virus may be of dietary origin, although this will need to be resolved with additional sampling.

The *Hepeviridae* is a family of positive-sense, single-stranded RNA viruses that infect vertebrates. We identified six novel hepevirus species in the Eurasian blackbird (*n* = 1), bellbird (*n* = 1), song thrush (*n* = 1), dunnock (*n* = 1) and silvereye (*n* = 2) libraries (Figure 9). All of these viruses fell into a clade primarily comprising viruses identified in metagenomic studies from a variety of hosts and environments, making their true hosts difficult to determine. However, our viruses did not group together within this clade, falling into three distinct clusters. Avian associated hepe-like virus 2 and 3 cluster (although with weak support) with sogatella furcifera hepe-like virus found in a plant-hopper, Barns Ness breadcrumb sponge hepe-like virus 4, and “Hepeviridae sp.” found in birds. Avian associated hepe-like virus 1 clusters with three viruses found in the breadcrumb sponge *Halichondria panicea*, and forsythia suspensa hepe-like virus found in weeping forsythia. Finally, avian-associated hepe-like viruses 4, 5, and 6 cluster with French Guiana hepe-like virus [23] from a passerine (rufous-throated antbird, *Gymnopithys rufigula*), and “Hepeviridae sp.” also found in birds, possibly suggesting these viruses are avian. However, this clade also includes Hubei hepe-like virus 3 [55] found in a centipede.

The *Picornaviridae* is a family of positive-sense, single-stranded RNA viruses associated with both vertebrates and invertebrates. We identified 11 novel picorna-like virus species in the robin (*n* = 1), song thrush (*n* = 5), Eurasian blackbird (*n* = 4), dunnock (*n* = 1) and bellbird (*n* = 1) libraries (Figure 10). One virus (avian associated picorna-like virus 8) was found in both the Eurasian blackbird and song thrush libraries, with 100% amino acid identity. All of the viruses identified fell into a ‘picorna-like’ clade, distinct from other currently described genera. This clade includes viruses that have been found in invertebrates, plants, fungi, and vertebrates, such that it is challenging to identify the true host of these viruses. Avian-associated picorna-like viruses 7 and 8 are most closely related to viruses found in vertebrates (bat picornaviruses and Picornaviridae sp. found in birds), although these sit within a larger (poorly supported) clade of predominately invertebrate viruses. Avian associated picorna-like viruses 1, 9 and 10 sit within a clade of predominately arthropod viruses. Avian-associated picorna-like viruses 6 and 11 sit within a mixed clade containing viruses found in vertebrates and invertebrates although with relatively long branch lengths. However, avian associated picorna-like viruses 2–5 (found in the South Island robin and song thrush) group with nine other viruses found in vertebrates including five bird species, which tentatively suggests that these avian associated picorna-like viruses may be infecting the host birds rather than invertebrates or plants from their diet.

The *Reoviridae* is a family of double-stranded RNA viruses that infects a wide range of hosts including plants, invertebrates, and vertebrates. We identified seven reoviruses in the Eurasian blackbird (*n* = 2), fantail (*n* = 1), dunnock (*n* = 1), robin (*n* = 2) and silvereye (*n* = 1) libraries (Figure 11). Two viruses identified in the robin library (avian associated reo-like viruses 5 and 6) fell into the genus *Cypovirus* and are most closely related to the Soudat virus (46.5 and 54% amino acid identity). As cypoviruses have exclusively been isolated in insects, these two viruses are highly likely to be diet-related. Avian associated orbivirus 1 found in the silvereye library falls into the genus *Orbivirus*. Orbiviruses infect a wide range of invertebrate and vertebrate hosts, such that identifying the true host of avian-associated orbivirus 1 is challenging. However, it appears to fall into a clade of viruses that have been found to be exclusively associated with arthropods and is most closely related to anopheles hinesorum orbivirus found in mosquitos (76% amino acid identity), suggesting it is also diet-related. Avian associated reo-like virus 1 and 2 both found in the Eurasian blackbird library fall closest to the *Phytoreovirus* genus which infects plants, suggesting it may also be diet-related. The remaining two reoviruses we identified (avian-associated reo-like viruses 3 and 4) do not fall close to any currently described genus, making identifying their true host difficult.

## 4. Discussion

We provide the first characterization of the RNA viromes of New Zealand passerines, utilising data obtained from the cloacal swabs of birds showing no obvious clinical signs of disease from 12 New Zealand bird species. From this metagenomic analysis, we found a total of 470 different virus species from 43 viral families, of which 436 viruses from 36 families were assumed to be of dietary/microbiome origin as they belonged to families that exclusively infect plant, fungi, or microorganisms. Phylogenetic analysis of the remaining 34 viruses from seven virus families indicated that only three of these are highly likely to replicate in avian hosts with the remainder more likely diet-associated. We also observed distinct and significant differences in the overall viral diversity according to the predominant diet of the host species, although this needs to be verified using studies with larger sample sizes.

The three novel viruses likely to infect their avian hosts were alphaherpesvirus, siadenovirus, and avastrovirus in the song thrush, Eurasian blackbird, and silvereye library, respectively. Turdid alphaherpesvirus 1 was most closely related (81.5% amino acid identity) to psittacid alphaherpesvirus 1 which causes Pacheco’s disease, a fatal respiratory disease in parrots [56]. Other viruses in the genus *Alphaherpesvirus* include gallid alphaherpesvirus 1 (associated with avian infectious laryngotracheitis, a major disease in domestic fowl) [56], and passerid herpesvirus 1 found in Gouldian finches (*Chloebia gouldiae*), which also causes fatal respiratory disease [57]. Unfortunately, as there was no capsid sequence available for passerid herpesvirus 1 it could not be included in our phylogenetic analysis. Turdid alphaherpesvirus 1 is the first iltovirus to be discovered in passerines since passerid herpesvirus 1 was identified in 2003. The song thrushes sampled here had no overt signs of disease, although given the latent nature of herpesviruses disease-associations are difficult to draw and there is potential for disease in other bird species following cross-species transmission. Herpesviruses are considered predominantly host-specific with potential virus-host co-divergence on time-scales of many millions of years [58]. However, recent analyses have shown that some herpesviruses can jump species barriers and establish endemic infections in non-definitive hosts, often causing disease and mortality [58]. Although rarely directly observed, there are examples of cross-species virus transmission between bird species: columbid herpesvirus 1 has been found in pigeons, falcons, and owls, with transmission thought to have occurred through predator-prey interactions [59]. Psitaccid alphaherpesvirus 1 also has a wide host range among parrots [60], suggesting frequent cross-species transmission among these related species.

We also identified a novel siadenovirus in the Eurasian blackbird library, most closely related to myna siadenovirus found in Australia [61]. Recent studies have identified siadenoviruses in multiple passerine species both in the wild and in captivity, including finches, silvereyes, and mynas [61,62,63]. To our knowledge, this is the first siadenovirus identified in a Eurasian blackbird, and the first siadenovirus identified in New Zealand. This recent discovery of siadenovirus diversity in multiple passerine species in Australia, Europe, and now New Zealand suggests that passerines might be an important host of siadenoviruses globally, not apparent until better sampling in recent years, and that there are likely to be additional siadenoviruses in New Zealand. Siadenoviruses are thought to be particularly adept at host switching, as indicated by the loose cluster of viruses from passerine, psittacid, raptor, and penguin showing a lack of host clustering in this genus [63]. Blackbird siadenovirus falls into a clade with four other passerine viruses, found in the common myna (*Acridotheres tristis*) and silvereye (*Zosterops lateralis*), both abundant in New Zealand, as well as the great tit (*Parus major*) and double-barred finch (*Taeniopygia bichenovii*) which are not present in New Zealand. This is possibly indicative of some virus-host co-divergence, although many other passerine viruses are also found across the genus. Passerine siadenoviruses also appear to have a broad host range [63], such that they may circulate through passerine communities, frequently jumping hosts. Many virus species within the genus *Siadenovirus* are either known or suspected to cause disease [64,65,66,67,68], and it will be important to ascertain whether this is associated with host jumping [63]. Adenoviruses are also known to be opportunistic pathogens in people, domestic animals, and wildlife [62,69,70,71]. The Eurasian blackbirds sampled here did not show obvious physical signs of disease, but as discussed with the alphaherpesvirus above, there is potential for disease in other passerine species if cross-species transmission were to occur.

A novel avastrovirus was identified in the silvereye library, most closely related to avian astrovirus found in another Zosteropidae (white-eye) sp. sample from Hong Kong. This virus sits in a passerine clade within the genus *Avastrovirus*, again indicative of some host-virus co-divergence. This clade was first discovered in French Guiana passerines in 2019 [21]. At the time, there was insufficient data to determine whether this clade was specific to this locality, and since this time additional viruses from this clade have been found in Hong Kong in multiple passerine species, and now in New Zealand. Accordingly, there is increasing evidence that passerines are important astrovirus hosts worldwide.

We also found a number of viruses where the true host could not be determined. In particular for viral families that infect both vertebrates and invertebrates (such as the *Picornaviridae*), identifying the true host can be challenging when the virus in question does not fall into an established genus. Importantly, even in viral families that are assumed to only infect vertebrates, metagenomic studies have found viruses belonging to these families in invertebrates—such as the *Caliciviridae* and *Hepeviridae* [53,72]. In the case of these families, the viruses identified here fell into a clade with other viruses from metagenomic studies (including those found in invertebrates), which makes identifying their true host difficult. Some of these viruses were closely related to other viruses found in the metagenomic analysis of bird cloacal swabs, although it is possible these are also diet-related. This highlights the challenge of determining the true origin of metagenomic sequences, particularly when viruses from widely different hosts group together in phylogenetic trees, and also highlights the importance of continuing to fill gaps in undersampled species and environments. Collecting tissue samples rather than cloacal swabs would help to confirm whether viruses are infecting the host, although these samples are harder to obtain. Sampling tissue from birds that had died (either from infectious disease or other causes) would help restrict the viruses detected to those more likely to directly infect the bird in question. However, any viruses identified could not be assumed to have caused disease or contributed to host death, such that their disease phenotype would still be difficult to assess.

The three species of virus that are highly likely to be avian viruses were all found in non-endemic species—the Eurasian blackbird, song thrush, and the silvereye. Due to the lack of passerine sampling worldwide, it is not possible to ascertain whether they brought these viruses with them when they were first introduced to New Zealand. The Eurasian blackbird and song thrush libraries also had a high diversity of viruses of unknown origin, particularly picorna-like viruses. In a concurrent study sampling the same individual birds they were also the only species to test positive for avian malaria [73], suggesting further study should be conducted to reveal the impact of avian malaria on viral diversity and abundance. However, statistical analysis indicated that whether birds had an endemic or introduced origin did not significantly impact the overall sampled viral community, although this may reflect our small sample size. Rather, the diet of the host was a significant factor when comparing all viruses found (i.e., both dietary and putative avian viruses), such that predominantly insectivorous birds had more diverse viromes than birds with other diets. Invertebrates have a high diversity and abundance of viruses [55], so it is unsurprising that their predators similarly have a high diversity of viruses—most of which will be from their diet and will not directly infect the host. However, our PERMANOVA analysis reveals that birds with similar diets did not have more viruses in common, despite having similar levels of diversity. Indeed, most of the bird species we sampled have highly varied diets (particularly the insectivores), which likely explains the lack of similarity between the host viromes. However, a larger study with more host species of each diet type is required to confirm the trends we detected in this study.

Finally, we found no evidence of virus transmission from introduced to endemic species, although given the size of our study and the lack of previous sampling in New Zealand passerines it is unlikely we would have detected this. Increased sampling and surveillance of New Zealand avifauna, along with the wider ecological communities they inhabit, would greatly improve our knowledge of the viral ecology of New Zealand wildlife, and global viral patterns.

## Figures and Tables

**Figure 1 viruses-14-01364-f001:**
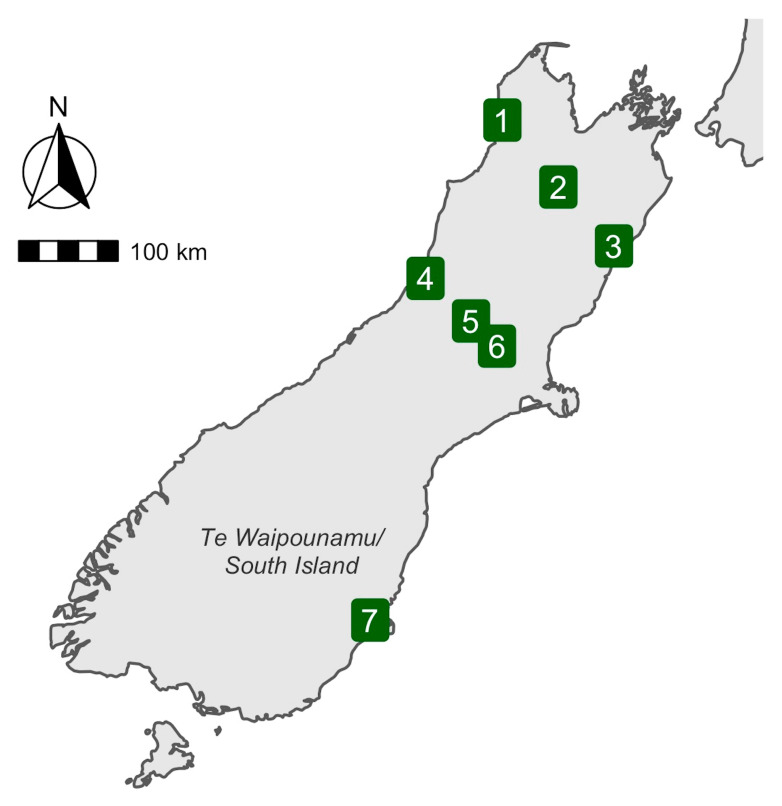
Map of the South Island/Te Waipounamu, New Zealand. Location of mist net sites shown as numbered green squares.

**Figure 2 viruses-14-01364-f002:**
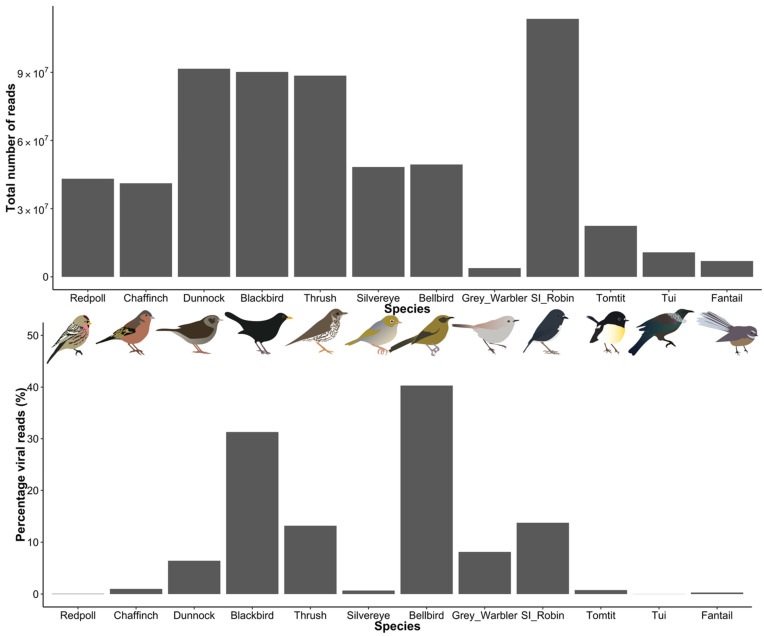
Read counts and the percentage of viral reads (%) for libraries from 12 New Zealand bird species. Bird images by Michelle Wille.

**Figure 3 viruses-14-01364-f003:**
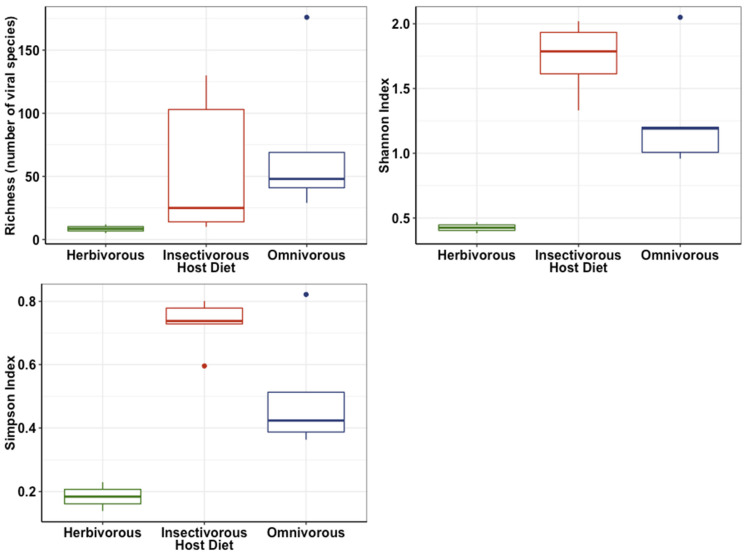
Boxplots showing the differences in Richness (number of viral species) (**top left**), Shannon index (**top right**) and Simpson index (**bottom left**) between hosts with different diets.

**Figure 4 viruses-14-01364-f004:**
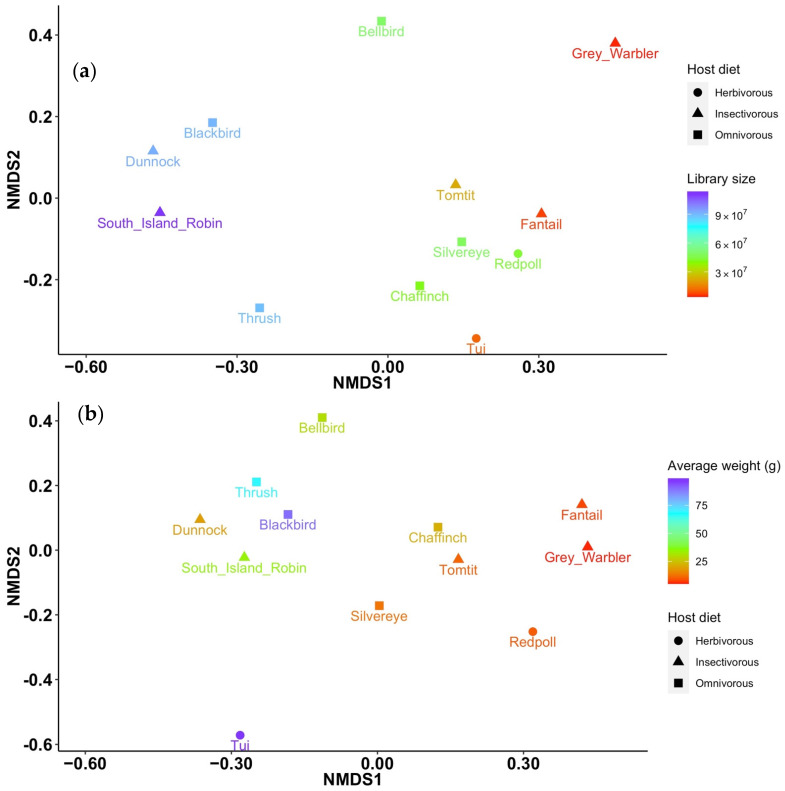
Non-metric multi-dimensional scaling plots applying the Bray–Curtis dissimilarity matrix for viral abundance and diversity, showing (**a**) the relative similarity/differences in the viral community at the virus species level in each host species (stress = 0.13), with the total number of reads for each library shown using the colour gradient and the host diet indicated using symbols, and (**b**) the relative similarity/differences in the viral community at the virus genus level in each host species (stress = 0.09), with the average weight in grams of the host species shown using the colour gradient and the host diet shown using symbols.

**Figure 5 viruses-14-01364-f005:**
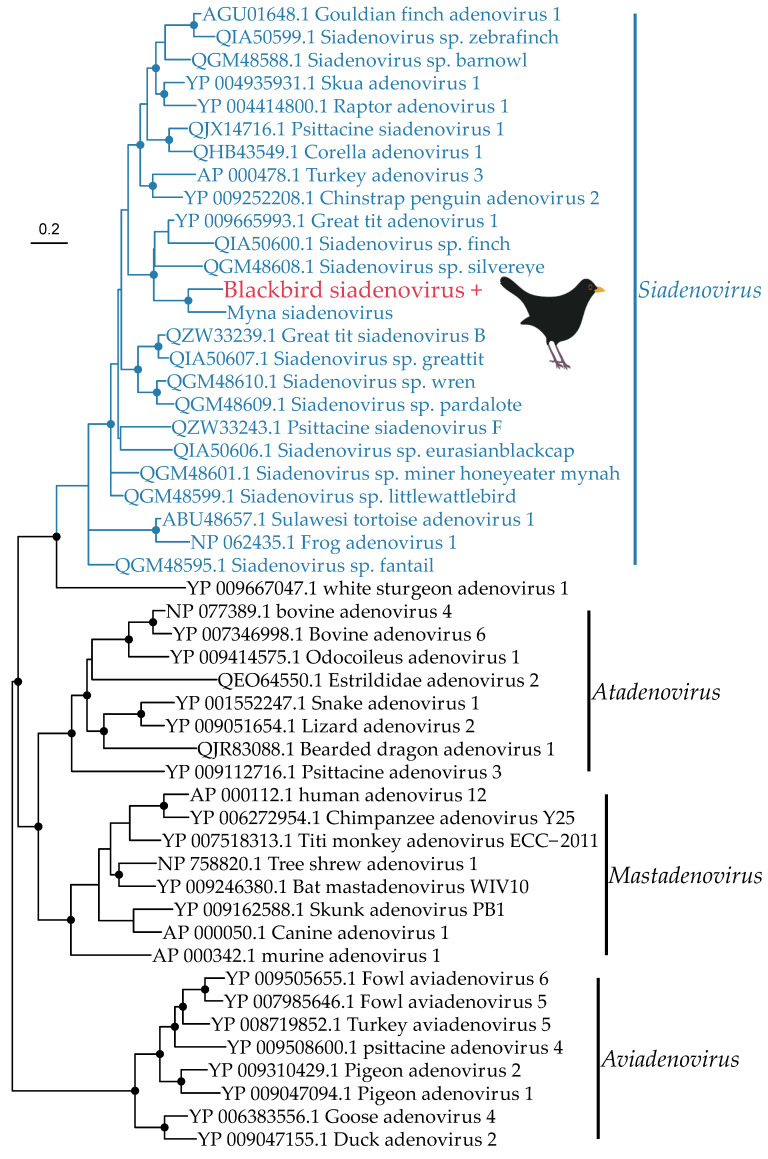
Phylogeny of the *Adenoviridae* based on the DNA polymerase gene (alignment length of 773 amino acids). The virus from this study (blackbird siadenovirus) has a ‘+’ after the name and is shown in red, and the genus that this virus belongs to (*Siadenovirus)* is shown in blue. Related viruses and their genera are shown in black. Black circles on nodes show bootstrap support values of more than 90%. Branches are scaled according to the number of amino acid substitutions per site, shown in the scale bar. The tree is midpoint rooted. Great tit siadenovirus B and psittacine siadenovirus F have not been designated a species classification by ICTV at the time of publication. Bird images in all tree figures by Michelle Wille.

**Figure 6 viruses-14-01364-f006:**
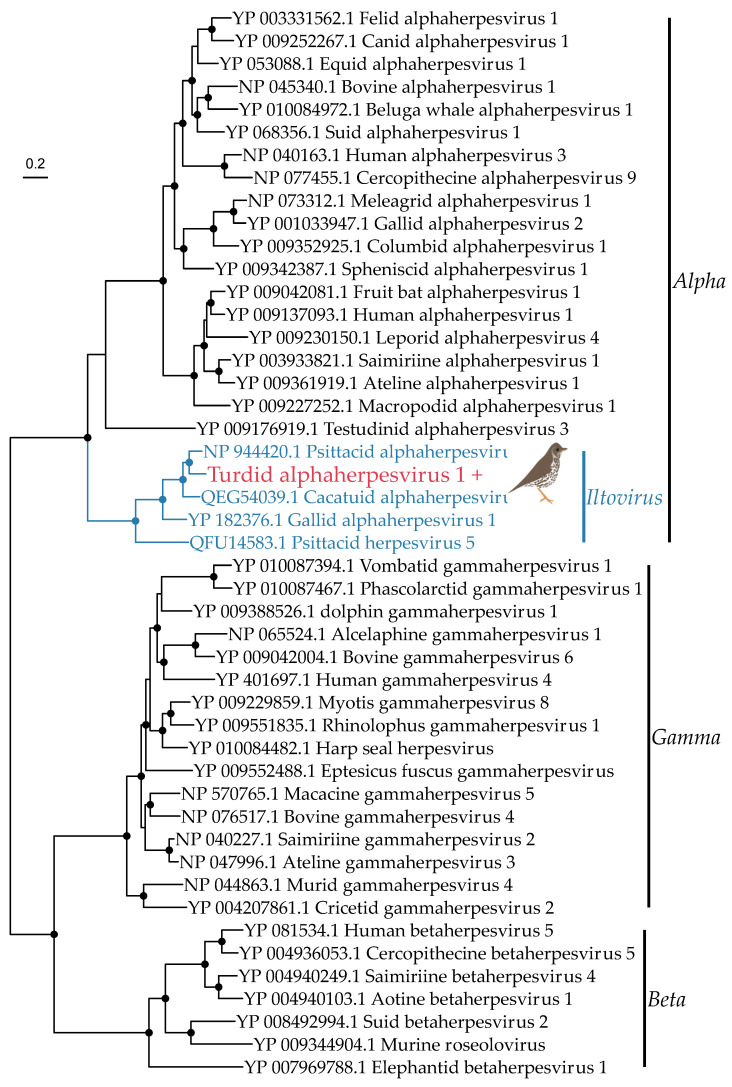
Phylogeny of the *Herpesviridae* based on the capsid gene (alignment length of 1279 amino acids). The *alpha*-, *beta*- and *gammaherpesvirinae* subfamilies are shown in black. Colours and symbols as per Figure 5.

**Figure 7 viruses-14-01364-f007:**
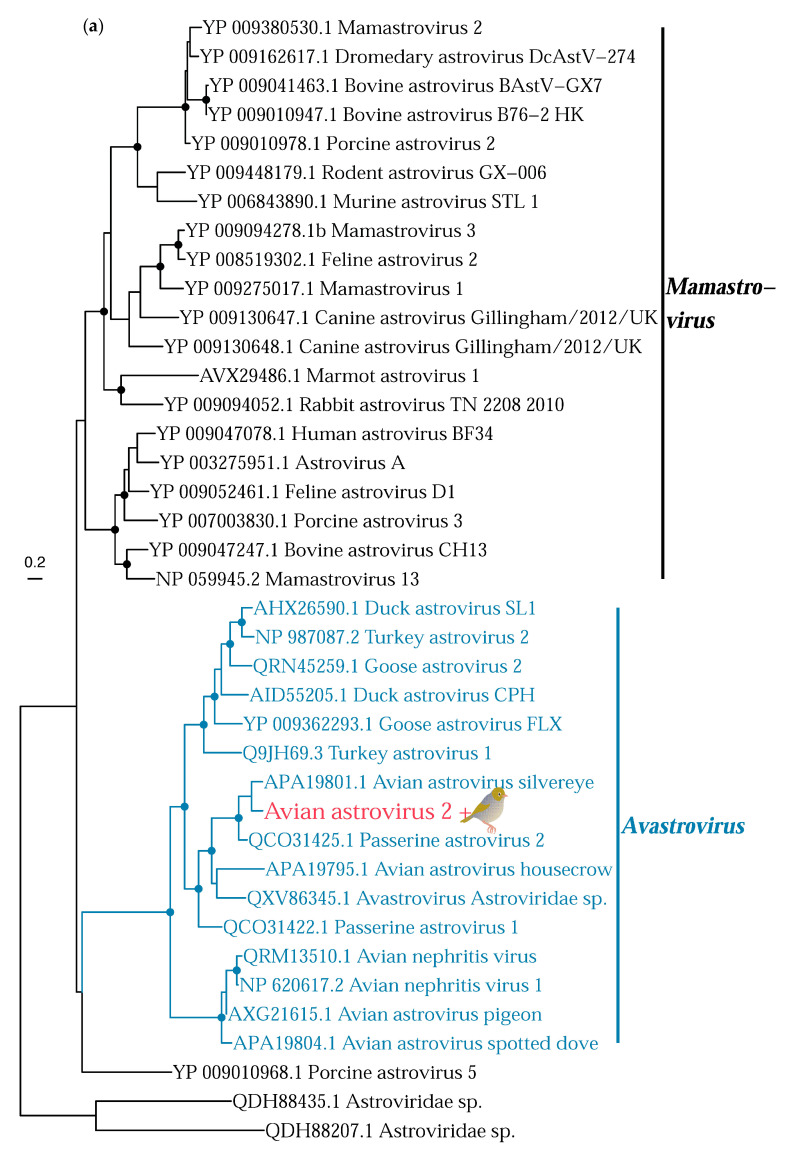
(**a**) Phylogeny of the *Astroviridae,* alignment length of 1306 amino acids. (**b**) Phylogeny of the bastrovirus clade, alignment length of 1305 amino acids. Both phylogenies are based on the RNA polymerase gene. Colours and symbols as shown in Figure 5.

**Figure 8 viruses-14-01364-f008:**
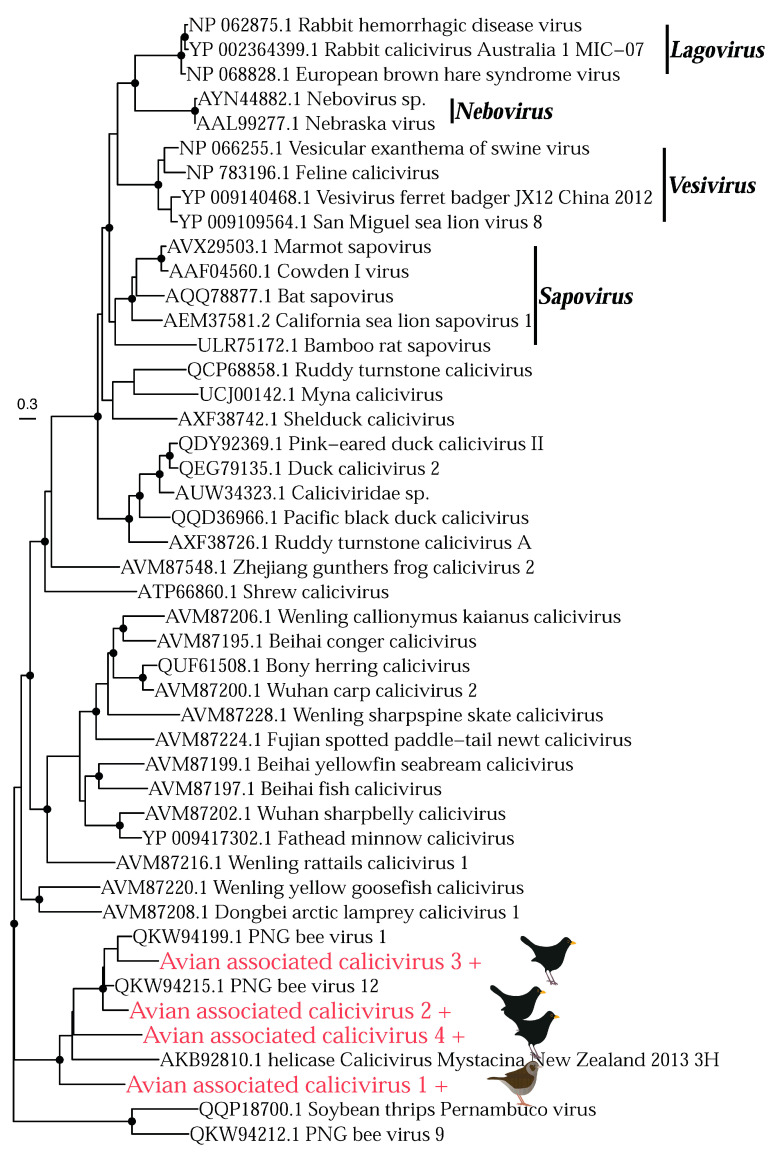
Phylogeny of the *Caliciviridae* based on the non-structural polyprotein (alignment length of 1087 amino acids). Key genera are shown in black with a black bar; all viruses below bamboo rat sapovirus on the phylogeny have not currently been classified into a genus by ICTV. Colours and symbols as shown in Figure 5.

**Figure 9 viruses-14-01364-f009:**
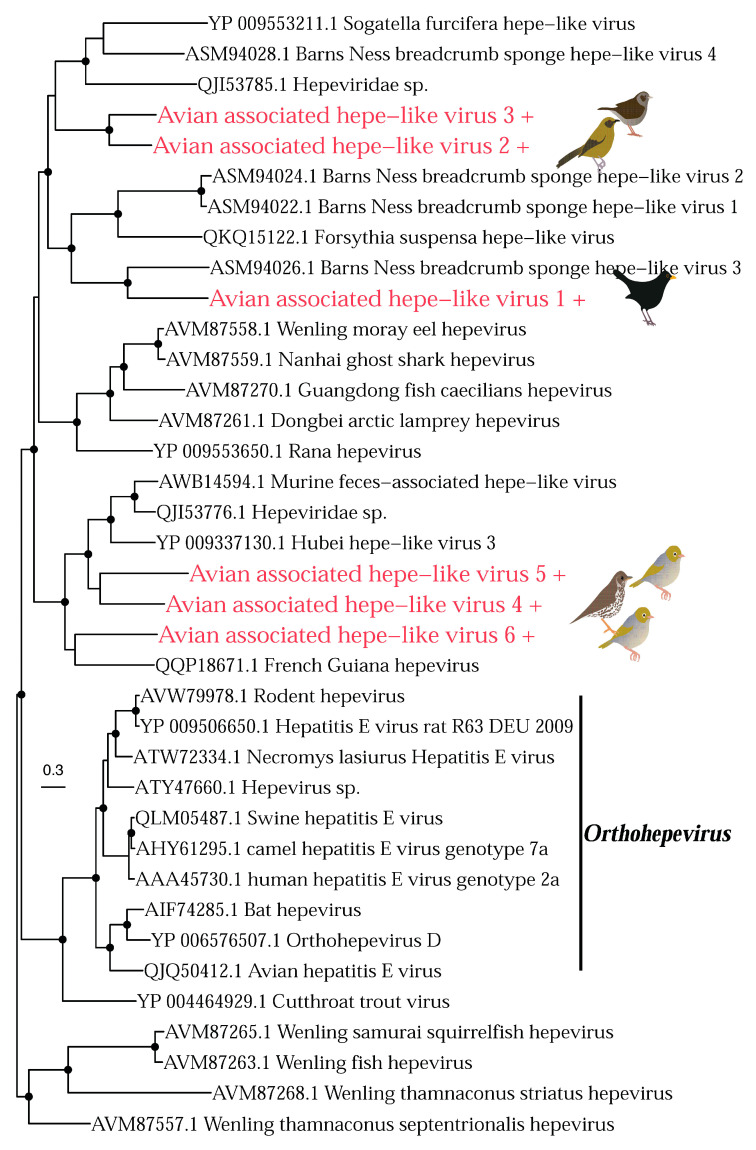
Phylogeny of the *Hepeviridae* based on the non-structural polyprotein (alignment length of 840 amino acids). Colours and symbols as shown in Figure 5.

**Figure 10 viruses-14-01364-f010:**
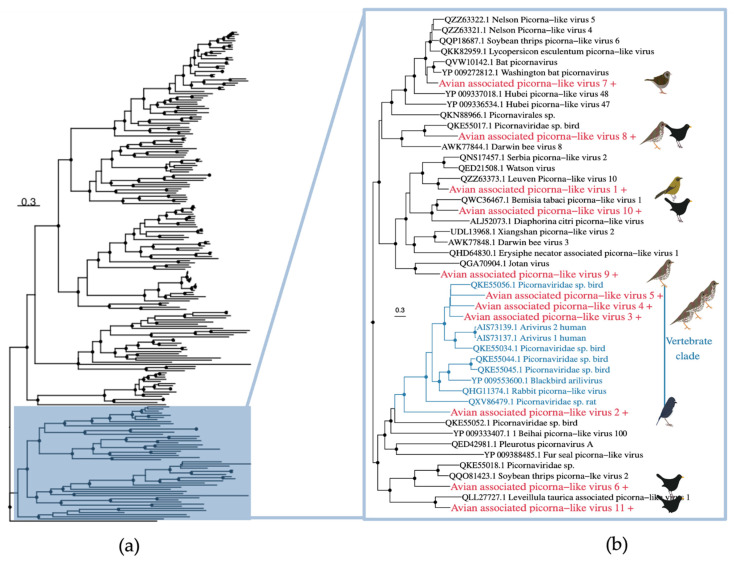
(**a**) Phylogeny of *Picornaviridae* based on the RNA-dependant RNA polymerase gene (alignment length of 2139 amino acids). (**b**) Magnification of a clade containing the viruses from this study. A clade of vertebrate viruses is shown in blue. Colours and symbols as shown in Figure 5.

**Figure 11 viruses-14-01364-f011:**
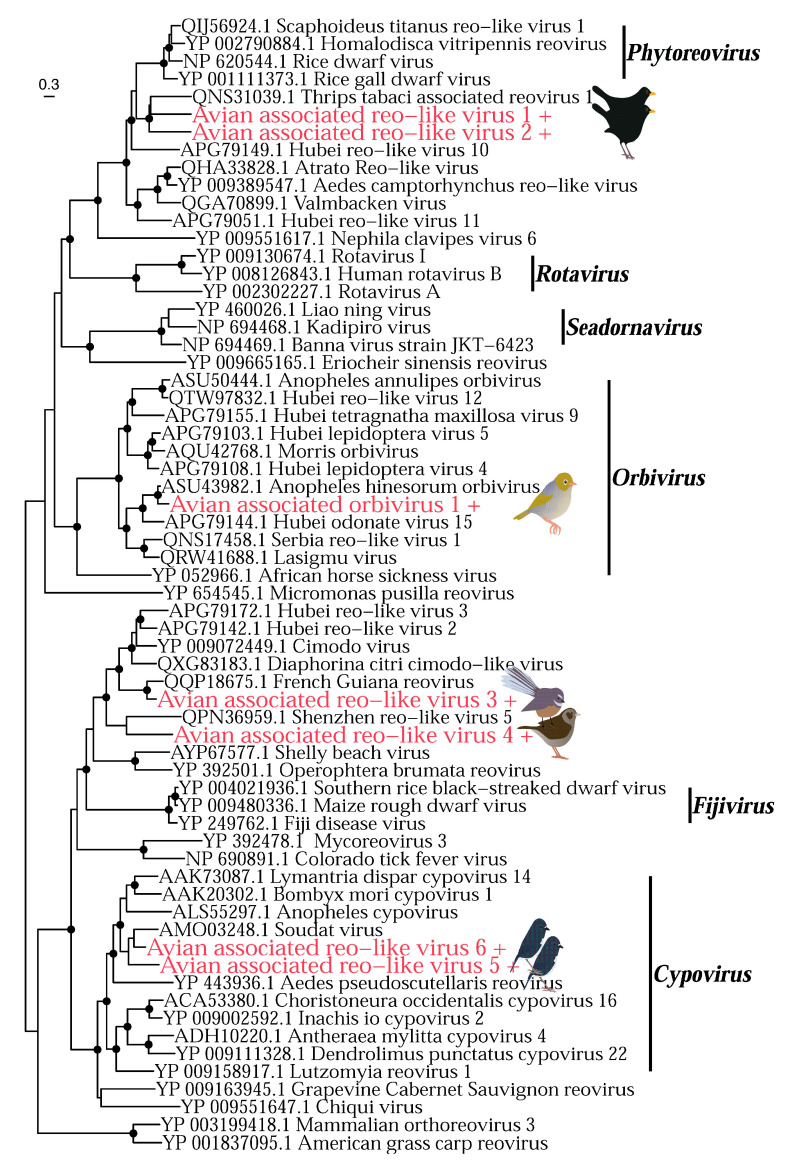
Phylogeny of the *Reoviridae* based on the RNA-dependent RNA polymerase gene (alignment length of 684 amino acids). Colours and symbols as shown in Figure 5.

**Table 1 viruses-14-01364-t001:** Summary of species sampled, including the common and Latin names, the origin of the species in New Zealand, the number of individual birds sampled, and the predominant diet of each species. Endemic = evolved in New Zealand, Introduced = evolved elsewhere and established in New Zealand via either deliberate human introduction or natural dispersal.

Common Name(s)	Scientific Name	Species Origin	Number Sampled	Predominant Diet
Common redpoll	*Carduelis flammea*	Introduced	4	Herbivorous
Chaffinch	*Fringilla coelebs*	Introduced	5	Omnivorous
Dunnock	*Prunella modularis*	Introduced	4	Insectivorous
Eurasian blackbird	*Turdus merula*	Introduced	4	Omnivorous
Song thrush	*Turdus philomelos*	Introduced	3	Omnivorous
Silvereye/tauhou	*Zosterops lateralis*	Introduced	11	Omnivorous
Bellbird/korimako	*Anthornis melanura*	Endemic	12	Omnivorous
Grey warbler/riroriro	*Gerygone igata*	Endemic	2	Insectivorous
South Island robin/kakaruai	*Petroica australis*	Endemic	4	Insectivorous
Tomtit/miromiro	*Petroica macrocephala*	Endemic	4	Insectivorous
Tūī	*Prosthemadera novaeseelandiae*	Endemic	3	Herbivorous
Fantail/pīwakawaka	*Rhipidura fuliginosa*	Endemic	3	Insectivorous

**Table 2 viruses-14-01364-t002:** The alpha diversity of each species/library—the richness (number of species), Shannon index and Simpson index, and the predominant diet of each species.

Species/Library	Richness	Shannon	Simpson	Predominant Diet
Bellbird	48	0.95839607	0.36300495	Omnivorous
Common redpoll	12	0.38264421	0.13870225	Herbivorous
Chaffinch	29	1.00701147	0.38721558	Omnivorous
Grey warbler	14	1.78694747	0.80119441	Insectivorous
South Island robin	103	1.33109767	0.59564466	Insectivorous
Tomtit	25	1.93339985	0.72866524	Insectivorous
Dunnock	130	2.02011625	0.77871149	Insectivorous
Tūī	5	0.46803031	0.2296815	Herbivorous
Fantail	10	1.61362017	0.73783657	Insectivorous
Eurasian blackbird	176	1.1909211	0.51274955	Omnivorous
Song thrush	69	2.04993336	0.82176403	Omnivorous
Silvereye	41	1.20118728	0.42349987	Omnivorous

**Table 3 viruses-14-01364-t003:** The details of each virus identified—the viral family, host species, the abundance, expressed as reads per million (RPM), the gene used in the phylogenetic trees, and the length in amino acids of that gene sequence. RdRp = RNA dependent RNA polymerase.

Virus	Viral Family	Host Species	Abundance (RPM)	Gene	Length (Amino Acids)
Blackbird siadenovirus	*Adenoviridae*	Blackbird	21.45	DNA polymerase	130
Avian astrovirus 2	*Astroviridae*	Silvereye	6.6	RdRp	211
Nelson astrovirus-like 1	*Astroviridae*	Robin	40115.8	RdRp	1402
		Tūī	0.93	RdRp	104
Avian associated bastrovirus 1	*Astroviridae*	Tomtit	314.9	RdRp	134
Avian associated bastrovirus 2	*Astroviridae*	Thrush	8956.92	RdRp	834
Avian associated hepe-like virus 1	*Hepeviridae*	Blackbird	2.93	Replicase	205
Avian associated hepe-like virus 2	*Hepeviridae*	Bellbird	20.33	Replicase	105
Avian associated hepe-like virus 3	*Hepeviridae*	Dunnock	228.58	Replicase	1350
Avian associated hepe-like virus 4	*Hepeviridae*	Thrush	170.6	Replicase	1760
Avian associated hepe-like virus 5	*Hepeviridae*	Silvereye	26.99	Replicase	126
Avian associated hepe-like virus 6	*Hepeviridae*	Silvereye	14.62	Replicase	109
Turdid alphaherpesvirus 1	*Herpesviridae*	Thrush	1207.78	Major capsid protein	1459
Avian associated calicivirus 1	*Caliciviridae*	Dunnock	417.46	RdRp	791
Avian associated calicivirus 2	*Caliciviridae*	Blackbird	0.75	RdRp	145
Avian associated calicivirus 3	*Caliciviridae*	Blackbird	0.77	RdRp	142
Avian associated calicivirus 4	*Caliciviridae*	Blackbird	3.12	RdRp	148
Avian associated picorna-like virus 1	*Picornaviridae*	Bellbird	16.34	RdRp	141
Avian associated picorna-like virus 2	*Picornaviridae*	Robin	5.04	RdRp	193
Avian associated picorna-like virus 3	*Picornaviridae*	Thrush	30.64	RdRp	281
Avian associated picorna-like virus 4	*Picornaviridae*	Thrush	54.1	RdRp	360
Avian associated picorna-like virus 5	*Picornaviridae*	Thrush	27.29	RdRp	407
Avian associated picorna-like virus 6	*Picornaviridae*	Blackbird	1.03	RdRp	101
Avian associated picorna-like virus 7	*Picornaviridae*	Dunnock	17.32	RdRp	186
Avian associated picorna-like virus 8	*Picornaviridae*	Blackbird	1.73	RdRp	173
		Thrush	250.73	RdRp	722
Avian associated picorna-like virus 9	*Picornaviridae*	Thrush	6.96	RdRp	181
Avian associated picorna-like virus 10	*Picornaviridae*	Blackbird	829.51	RdRp	1708
Avian associated picorna-like virus 11	*Picornaviridae*	Blackbird	0.43	RdRp	130
Avian associated orbivirus 1	*Reoviridae*	Silvereye	19.36	RdRp	142
Avian associated reo-like virus 1	*Reoviridae*	Blackbird	5.37	RdRp	304
Avian associated reo-like virus 2	*Reoviridae*	Blackbird	16.82	RdRp	897
Avian associated reo-like virus 3	*Reoviridae*	Fantail	90.34	RdRp	113
Avian associated reo-like virus 4	*Reoviridae*	Dunnock	5.84	RdRp	73
Avian associated reo-like virus 5	*Reoviridae*	Robin	2.33	RdRp	161
Avian associated reo-like virus 6	*Reoviridae*	Robin	1.48	RdRp	85

## Data Availability

Sequencing data has been deposited in the Sequence Read Archive (SRA) with the following accession numbers SAMN27393654-65. The consensus sequences of all novel viruses have been submitted to GenBank and assigned accession numbers ON304002-41.

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
