# Peer review of "Metatranscriptomic Comparison of Viromes in Endemic and Introduced Passerines in New Zealand"

_viruses, 2022, doi:10.3390/v14071364_

Round 1

Reviewer 1 Report

French et al. describe a large study on the metatranscriptomic comparison of viromes in passerines in New Zealand. The used metatransciptomics proved to be useful to find a lot of RNA viruses but was not really effective to gain longer sequences from DNA viruses (only very short fragments were recovered). The work is interesting from the point of virus hunting and gives useful data for the study of the diversity of several virus families. However, it has some limitations in the characterization of some of the identified viruses where the available sequences are rather short. Another problem is with the sampling. As cloacal samples were studied, it is a continuous problem to guess if the found virus is a real avian virus (replicating in the host) or just have a dietary origin. The authors try to make an intelligent guess about these by comparing the gained sequences to earlier findings, but obviously this huge work would give more precise results (especially in the case of the RNA viruses) if bird organs were used (e.g from naturally died /and thus supposedly ill/ birds). The work is slightly Australia and New Zealand concentrated when citing the earlier found avian viruses. The text is a good description of the work but certainly would benefit from being shortened. E.g., the legend for the seven figures with phylogenetic trees repeats several information seven times. Or the very end (the last sentence) of the Discussion what advantage could a larger sampling mean is unnecessary as there is actually a “large-scale undersampling” here.

A general problem is that the authors apparently mix the meaning of a species (a theoretical idea about grouping some similar viruses under a taxonomical name) with the actual virus (a detected/sequenced “real” virus). A species has no sequence only a MEMBER of a species has a sequence. The species names must be written in italics and starting with capital. The virus names should never be in italics and starting with a capital (except if it is a geographical or personal name, or if it is the start of a sentence). Mixing of the species and virus names are very apparent in Figure 5, where the species names end with letters and the virus types/strains end with Arabic numbers, and they are continuously mixed as they were equal.

Some minor notes:

Line 280: The Adenoviridae [later Herpesviridae] is a family … that infects [it is not equivalent with “adenoviruses” but a single taxon, a family].

Table 3: Mistyping: “Adenovirirdae”

314: turdid alphaherpesvirus 1

316: alpha, beta- and gammaherpesvirinae [missing comma after alpha]

324. Correctly: 3.1.2. RNA viruses

326: infect mainly mammals and birds, but also other vertebrates

328: avian astrovirus 2

340: avian associated bastrovirus 2

386: avian associated hepe-like virus 4, 5 [with space]

426: Anopheles hinesorum orbivirus [not in italics]

467, 479: If you write passerid herpesvirus 1 why would be the columbid herpesvirus 1 with hyphen.

509: in another Zosteropidae (white-eye) sp. [a single species, no plurals]

578: ON304002-41 are not released yet. If you wish to keep confidential your sequences, at least provide them to the reviewers

627: Astroviridae should be in italics

696: Mistyping: “Clin.l”

714: Mistyping: “J. Virol.l”

723: Pasteurella pneumotropica should be in italics

Author Response

French et al. describe a large study on the metatranscriptomic comparison of viromes in passerines in New Zealand. The used metatransciptomics proved to be useful to find a lot of RNA viruses but was not really effective to gain longer sequences from DNA viruses (only very short fragments were recovered). The work is interesting from the point of virus hunting and gives useful data for the study of the diversity of several virus families. However, it has some limitations in the characterization of some of the identified viruses where the available sequences are rather short.

Response: Thank you for your review. While many of the virus sequences we detected were short, which is usually the case for metranscriptomic sequencing, we did detect sequences encoding full length proteins, including the full-length capsid gene for Turdid alphaherpesvirus 1 which is a DNA virus.

Another problem is with the sampling. As cloacal samples were studied, it is a continuous problem to guess if the found virus is a real avian virus (replicating in the host) or just have a dietary origin. The authors try to make an intelligent guess about these by comparing the gained sequences to earlier findings, but obviously this huge work would give more precise results (especially in the case of the RNA viruses) if bird organs were used (e.g from naturally died /and thus supposedly ill/ birds).

Response: We agree that this is an important issue, and it is something that we deal with on a daily basis. We also believe that our analysis is better than an intelligent guess. We have however, added more on this point in the discussion, lines 544-548

The work is slightly Australia and New Zealand concentrated when citing the earlier found avian viruses. The text is a good description of the work but certainly would benefit from being shortened. E.g., the legend for the seven figures with phylogenetic trees repeats several information seven times. Or the very end (the last sentence) of the Discussion what advantage could a larger sampling mean is unnecessary as there is actually a “large-scale undersampling” here.

Response: The text has been shortened, including the figure legends.

A general problem is that the authors apparently mix the meaning of a species (a theoretical idea about grouping some similar viruses under a taxonomical name) with the actual virus (a detected/sequenced “real” virus). A species has no sequence only a MEMBER of a species has a sequence. The species names must be written in italics and starting with capital. The virus names should never be in italics and starting with a capital (except if it is a geographical or personal name, or if it is the start of a sentence). Mixing of the species and virus names are very apparent in Figure 5, where the species names end with letters and the virus types/strains end with Arabic numbers, and they are continuously mixed as they were equal.

Response: Many thanks for pointing this out. This has been corrected throughout the paper, including Figure 5. Please note that great tit siadenovirus B and psittacine siadenovirus F are virus names despite ending with a letter: they currently have not been given a species classification by ICTV. Hence, there is no more we can do in these two cases.

Some minor notes:

Line 280: The Adenoviridae [later Herpesviridae] is a family … that infects [it is not equivalent with “adenoviruses” but a single taxon, a family].

Table 3: Mistyping: “Adenovirirdae”

314: turdid alphaherpesvirus 1

316: alpha, beta- and gammaherpesvirinae [missing comma after alpha]

  1. Correctly: 3.1.2. RNA viruses

326: infect mainly mammals and birds, but also other vertebrates

328: avian astrovirus 2

340: avian associated bastrovirus 2

386: avian associated hepe-like virus 4, 5 [with space]

426: Anopheles hinesorum orbivirus [not in italics]

467, 479: If you write passerid herpesvirus 1 why would be the columbid herpesvirus 1 with hyphen.

509: in another Zosteropidae (white-eye) sp. [a single species, no plurals]

578: ON304002-41 are not released yet. If you wish to keep confidential your sequences, at least provide them to the reviewers

627: Astroviridae should be in italics

696: Mistyping: “Clin.l”

714: Mistyping: “J. Virol.l”

723: Pasteurella pneumotropica should be in italics

Response: All minor comments have been addressed. Thanks for spotting these. The sequences were submitted to GenBank prior to the paper submission but are still being processed, so haven’t yet been released. We were certainly not intending to keep the sequences confidential and if the reviewers wish to see them we are happy to provide them.

Reviewer 2 Report

The article by French and colleagues presented the comparison of viromes in wild passerines in New Zealand. The methods are given in detail along with adequate sampling information. The analysis is also satisfactory, and the discussion is comprehensive.

Although the study and may be the analysis started prior to the recent taxonomy revision by ICTV, I would suggest using virus names in the manuscript and phylogeny according to the current virus nomenclature.

As an example, Skua adenovirus 1 as Skua siadenovirus A and Frog adenovirus 1 as Frog siadenovirus A.

And for me, information given in 533 -537 are mostly the background and might be better to present under introduction.

Author Response

The article by French and colleagues presented the comparison of viromes in wild passerines in New Zealand. The methods are given in detail along with adequate sampling information. The analysis is also satisfactory, and the discussion is comprehensive.

Response: Thank you for your review.

Although the study and may be the analysis started prior to the recent taxonomy revision by ICTV, I would suggest using virus names in the manuscript and phylogeny according to the current virus nomenclature.

As an example, Skua adenovirus 1 as Skua siadenovirus A and Frog adenovirus 1 as Frog siadenovirus A.

Response: In line with reviewer 1’s comments to be consistent with regards to virus species and virus names, we have changed all labels in Figure 5 to be virus names. Thus, we have kept skua adenovirus 1 and frog adenovirus 1, as that is the virus name.

And for me, information given in 533 -537 are mostly the background and might be better to present under introduction.

Response: We agree, and this has been moved to the introduction (lines 41-45).